# Study Confirms Safety and Effectiveness of Intra-Articular Glucocorticoids for Painful Hip Dislocation in Children and Young Adults with Neurologic Impairment

**DOI:** 10.3390/children10081353

**Published:** 2023-08-06

**Authors:** Simone Benvenuto, Egidio Barbi, Silvia Boaretto, Matteo Landolfo, Francesco Rispoli, Giorgio Cozzi, Marco Carbone

**Affiliations:** 1Department of Medicine, Surgery, and Health Sciences, University of Trieste, 34127 Trieste, Italy; simone.benvenuto@burlo.trieste.it (S.B.); silvia.boaretto8@gmail.com (S.B.); francesco.rispoli@outlook.com (F.R.); 2Institute for Maternal and Child Health IRCCS Burlo Garofolo, 34137 Trieste, Italy; giorgio.cozzi@burlo.trieste.it (G.C.); marco.carbone@burlo.trieste.it (M.C.); 3Medical Clinic, Cattinara Hospital, Azienda Sanitaria Universitaria Giuliano Isontina, 34149 Trieste, Italy; matteo.landolfo@asugi.sanita.fvg.it

**Keywords:** intra-articular glucocorticoids, hip dislocation, neurologic impairment, cerebral palsy, orthopedic surgery

## Abstract

Background: Hip dislocation is a common source of pain in children with neurologic impairment. When medical interventions fail, orthopedic surgery does not guarantee a definitive result as the displacement may continue postoperatively and a second operation is often required. Methods: Retrospective analysis of data regarding the safety and effectiveness of an intra-articular corticosteroid injection (IACI) in 11 patients, aged 15 ± 5 years old, collected through a telephonic questionnaire administered to parents. Results: 21 IACIs were performed, a mean number of 1.9 ± 1.5 times for each patient, at a mean age (of the first IACI) of 13.5 ± 5 years. According to the parents, the IACI significantly lowered the number of participants experiencing pain (82% reduction) and using analgesics (60% reduction). There was also a significant improvement in the children’s hip mobility (63% reduction in patients experiencing stiffness), decubitus (90% reduction in obligated positioning), behavior (80% reduction in lamenting or crying patients), sleep quality (87.5% reduction in patients awakening every night), and caregivers’ quality of life (91% reduction in worried parents). The mean reported duration of the IACIs’ benefit was 5.4 ± 2.4 months (range 1–9), with a positive correlation with the number of IACIs (r = 0.48; *p*-value = 0.04) and a negative correlation with the age at the first injection (r = −0.71; *p*-value = 0.02). The only reported adverse event was mild local swelling in one child. Conclusions: the IACI could represent a safe and effective intervention for painful hip dislocation, both before and after surgery, with a long-lasting benefit which seems to increase as multiple IACIs are performed.

## 1. Introduction

Children with neurologic impairment from several causes, such as genetic syndromes, neurodegenerative disorders, traumatic brain injury, epileptic syndromes, and cerebral palsy (the most frequent, with an incidence of up to three per 1000 live births in Europe [1]), experience pain more frequently than their healthy peers [2]. Their pain is frequently underrecognized and undertreated [3], mostly because of their inability to self-report it or their uncommon way of communicating it (i.e., pain behaviors such as “freezing”, self-injurious behaviors, or misleading laughter). However, untreated pain profoundly impairs their quality of life in terms of mood, social skills, sleep, and physical and cognitive abilities, globally increasing the limitations associated with their neurologic condition [4]. Finally, its impact increases caregivers’ worries and frustration.

The complexity of these children’s pain is further increased by its multiple possible etiologies. While accidental injury is a rare cause, iatrogenic pain (e.g., venipuncture) and the condition’s related morbidities are the most common source of pain. In fact, apart from common problems (otitis media, headache, menses, and appendicitis), these children often suffer from gastrointestinal (GI) disorders (constipation, gastroesophageal reflux, and GI motility disorders), neuromuscular and/or neurogenic disorders (spasticity, dystonia, and neuropathic pain), caries, renal and/or gallbladder stones, and orthopedic conditions [5].

Among the latter, which include osteopenia and osteoporosis, fractures, and scoliosis, hip displacement (HD) is the second most frequent deformity after equinus deformity of the foot [6]. In particular, the severity of HD is strictly correlated with the child’s motor impairment, which is universally classified according to a five-level ordinal scale, i.e., the Gross Motor Function Classification System (GMFCS) [7]. Overall, HD affects one-third of children with cerebral palsy (CP), reaching a 70–90% risk among those with the lowest functional level, expressed through the GMFCS level V [8]. Reduced hip motion impairs function and quality of life, and up to 70% of these hips become painful [9,10]. Indeed, hip displacement is the first cause of pain in outpatient children with CP [11].

At birth, children with neurologic impairment mostly present normal hip anatomy. In those affected, lateral displacement of the femoral head from the acetabulum usually progresses slowly, leading to hip subluxation and eventually dislocation. However, the exact pathogenetic mechanism of this process is not clear. Historically, HD has been considered the result of the altered tone of the muscles around the hip joint and, in particular, the spasticity in the hip adductors and flexors [12]. Recently, some studies focused on a possible role for abnormal femoral geometry, based on the high prevalence of HD among children with hypotonia, and the observed increase in femoral neck anteversion and neck shaft angle from GMFCS level I to V [13,14]. In fact, factors which determine this abnormal geometry remain to be established.

Clinically, HD may be identified through reduced hip mobility, pelvic obliquity, scoliosis, sitting imbalance, and/or pain at hip mobilization, which usually accompany loss of sitting and walking tolerance, difficulty with dressing and bathing, and sleep disturbance [15]. Diagnosis requires a single anterior-posterior pelvis radiograph in order to calculate migration percentage (MP, also referred to as Reimers’ migration index), which is universally accepted as the most reliable method for monitoring HD [16]. This method measures the percentage of the femur head that has already migrated over the lateral margin of the acetabulum [17]. Hip subluxation is defined by an MP above 30%, while dislocation occurs over 80%. Although dislocation implies the complete loss of the articular congruity, a dislocated hip may still be painful in its early phase because of the impingement between the femur head and the margin of the acetabulum, which damages the femoral articular cartilage. With subsequent hip displacement, the femur head migrates away from the acetabulum towards the ilium, and this painful conflict is usually resolved. While hips with an MP <40% may correct spontaneously [18], those >60% are expected to progress to dislocation [19]. Factors such as age and ambulatory ability have been recognized as predictors of the speed of HD progression; in a study [20], non-ambulatory children had a 12% annual increase in MP, compared to only 2% in children who could walk.

The current optimal management for neurologic HD is based on hip surveillance programs [21], which usually start at 1 to 2 years of age in order to detect HD early and to favor preventive and reconstructive procedures, such as adductor psoas release surgery, “guided growth” intervention, or bony reconstructive surgery (femoral varus derotation osteotomy, pelvic Dega osteotomy), over salvage surgery (Schanz osteotomy, Girdlestone, Castle resection, McHale procedure, proximal femur prosthetic interposition arthroplasty [22,23,24]) which represents the last option when permanent dislocation and irreversible femur head damage occur. Low quality evidence is available for less invasive preventive strategies, such as a botulinum neurotoxin A (BoNT-A) injection to the hip adductors, hip flexors and hamstrings, an intrathecal baclofen infusion, an obturator nerve block, selective dorsal rhizotomy, positioning devices, and complementary and alternative medicine approaches [25]. However, a recent randomized trial showed the preventive efficacy of a hip brace towards neurologic hip displacement progression [26].

Different strategies are currently adopted when the hips become painful, including postural management, medical interventions, and, eventually, orthopedic surgery. Twenty-four-hour postural management and the implementation of adequate supportive equipment, with an early involvement of orthopedic services, may be of help in terms of general comfort and spasticity control [27]. Medical interventions include the broad use of analgesics, such as paracetamol, non-steroidal anti-inflammatory drugs (NSAIDs), and opioids, along with tone-modifying drugs, such as oral or intrathecal baclofen [28] and BoNT-A [29], in order to reduce pain due to spasticity, and anticonvulsants, such as gabapentin, which aim to address neuropathic pain, a frequent result of chronic undertreated pain [30]. Unfortunately, when chronic pain is no longer adequately controlled by medical interventions, surgery does not guarantee a definitive result as displacement may continue postoperatively, although at a slower rate, and a second operation is often required [31]. Moreover, some of these children might not have access to these procedures because of factors such as fragility, limited life expectancy, or general anesthesia contraindications.

While an intra-articular corticosteroid injection (IACI) is a standard of care in inflammatory disorders, such as juvenile idiopathic arthritis (JIA), and inflammatory-degenerative processes, such as osteoarthritis [32], limited evidence is available for children with neurologic HD, in contrast with its common use in clinical practice.

To our knowledge, so far only one retrospective study [33] has investigated the effectiveness of an IACI in 15 children with neurologic HD. It showed that injections of betamethasone reduced pain and improved parents’ and caregivers’ satisfaction, with a benefit lasting an average of 4 months but decreasing during follow up.

The aim of the present study was to assess the safety and effectiveness of the IACI in the setting of neurologic HD in terms of pain control and improvement to patients’ and caregivers’ quality of life.

## 2. Materials and Methods

### 2.1. Data Collection

We retrospectively reviewed data on patients with neurologic impairment and painful HD who received at least one IACI in the Pediatric Orthopedics and Traumatology Unit of our Institute from January 2018 to March 2023. Inclusion criteria were (1) a diagnosis of neurologic impairment from any cause, (2) painful HD of any grade (MP > 30%), and (3) having received at least one IACI to treat hip pain. Patients receiving hip surgery or other procedures for hip pain (such as BoNT-A injections, intrathecal baclofen infusion, obturator nerve block, and selective dorsal rhizotomy) in the 3 months preceding the IACI were excluded. Demographic and clinical data and information regarding the IACI’s effectiveness and safety were collected in an anonymized database through a Likert scale-based telephonic questionnaire administered to parents. In particular, the questionnaire was built to investigate the IACI’s effectiveness in terms of the following: use, type and frequency of medical and non-medical therapies for pain; presence, frequency and intensity of pain episodes; changes in hip mobility; decubitus; behavior; sleep quality and appetite; number of emergency department admissions for pain; and changes in caregivers’ quality of life. The IACI’s safety was assessed through local adverse events, such as swelling, bruising, erythema, depigmentation, skin atrophy, or any systemic adverse event including Cushingoid habitus, an acne-like rash, and sleep disorders.

### 2.2. IACI Description

At our Institute, an IACI is performed by a pediatric orthopedic surgeon while procedural sedation is obtained with intravenous ketamine by a pediatrician with specific expertise (i.e., pediatric anesthesiologist or a specifically trained pediatrician who has performed at least 30 procedural sedations and an anesthesiologists’ driven training in pediatric airway management). The multidisciplinary team includes a radiologic technologist to provide fluoroscopic guidance (Figure 1).

Indication to give an IACI is provided on a case-by-case basis, and repeated in the same patient as needed, based on the recurrence of chronic pain. We based this approach on the model of IACI in juvenile arthritis, which can be repeated. The pediatric orthopedic surgeon and the pediatrician jointly decide to repeat the injection when the following criteria are met: other causes of pain are ruled out; there are convincing symptoms; the patient is not responding to oral treatment; and a physical evaluation clearly points at a hip related relapse of pain.

Both sterile methylprednisolone acetate 40 mg/mL and 1% mepivacaine solution for injection are prepared and handled in accordance with the manufacturer’s recommendations. The patient lies supine in a neutral position with extended legs, with his/her caregiver standing nearby in order to reduce anxiety. As the appropriate level of sedation is reached, the caregiver leaves the room, and the procedure starts as a sterile aseptic technique. Skin is disinfected with chlorhexidine solution. A 22G needle is inserted under fluoroscopic guidance through the anterior capsule of the hip joint. Then 3 mL of 1% mepivacaine are injected, followed by 1 mL (40 mg) of methylprednisolone acetate solution (total intracapsular volume 4 mL). The needle is then withdrawn, hemostasis ensured, and a dressing is put over the injection site. Finally, the patient is monitored until awake.

### 2.3. Statistical Analysis

Continuous data were presented as mean and standard deviation. Categorical variables were expressed as numbers and percentage, and compared using Fisher’s exact test. Numerical variables were compared using Student’s *t*-test or Mann-Whitney’s test, after checking for distribution normality through Shapiro-Wilk’s test. Correlation between parameters was investigated through Pearson’s or Spearman’s correlation tests, depending on the appropriateness for the data. Differences were considered significant for *p*-value < 0.05. Statistical analysis was performed using SPSS software version 26.0 (IBM). A power analysis was conducted using G*Power version 3.1 for sample size estimation, based on data from the only previous study on the topic [33]. The effect size for the pain score reduction and the benefit duration in this study were 4.8 points and 4 months, respectively. With a significance criterion of α = 0.05 and power = 0.80, the minimum sample size needed with this effect size was N = 5.

## 3. Results

### 3.1. Population

During the study period, 11 patients were treated. All families could be contacted and agreed to participate in the data collection.

We included 11 patients, 6 males and 5 females, aged between 3 and 23 years old (mean 15 ± 5), with neurologic impairment as a consequence of genetic syndromes (5 participants; 46%), cerebral palsy (4; 36%), or metabolic syndromes (2; 18%); the most common comorbidity was epilepsy (9; 82%). Their daily therapies included anticonvulsants (9, 82%), proton-pump inhibitors (7; 64%), analgesics (4; 36%), laxatives (3; 27%), and bisphosphonates (3; 27%). All the participants were classified into GMFCS level V and presented hip dislocation (Reimers’ migration index > 80%), with such grade of migration that a painful conflict between the femur head and the acetabulum was still recognizable. Nine patients (82%) had already received orthopedic surgery at the time of the first IACI, such as vertebral arthrodesis (8, 73%), proximal femur osteotomy (2, 18%), and adductor lengthening or tenotomy (2, 18%). Three participants (27%) had received painful hip-related surgical procedures (i.e., femur osteotomy, adductor lengthening, or tenotomy) an average of 8.7 ± 4.4 years before the first IACI. None received surgery or other painful HD-related procedures during the study period. The mean age at hip pain onset was 10 ± 5 years (range 3–19).

### 3.2. IACI

Overall, 21 IACIs were performed, as fluoroscopic-guided arthrocentesis and injection of methylprednisolone acetate 40 mg plus 3 mL of 1% mepivacaine, under procedural sedation with intravenous ketamine. All of the IACIs were performed at our Institute.

Six patients received only one IACI, three received two, only one received three and another received six IACIs, resulting in a mean of 1.9 ± 1.5 IACIs for each patient. The mean age at first IACI was 13.5 ± 5 years (range 3.4–20). The follow up time from the first IACI varied between 5 and 45 months (mean 27 ± 17). Table 1 shows the IACIs’ timeframe for each patient receiving more than one procedure.

The results regarding the effectiveness of each patient’s first IACI in our cohort are shown in Table 2.

An IACI significantly lowered the number of patients using analgesics (60% reduction), especially paracetamol, the most used, with reduced frequencies (all those requiring more than daily or 2–3 administrations per week switched to daily or weekly/occasional administration) and a five-fold increase in the rate of response to these drugs. As expected, no differences were found regarding physiotherapy, which all participants continued receiving as part of their therapeutic schedule. According to the parents, an IACI significantly reduced the number of patients experiencing pain (82% reduction), with decreased pain frequency and intensity for one of the two remaining “non-responders”. Of note, the only child who did not experience any benefit suffered from recurrent episodes of hip dislocation, which his parents had learned to reduce at home. Moreover, significant improvements in the patients’ hip mobility (63% reduction in patients experiencing stiffness), decubitus (90% reduction in obligated positioning), behavior (80% reduction in lamenting or crying patients), and sleep quality (87.5% reduction in patients awakening at least once a night) were reported. Caregivers’ quality of life significantly improved as well, with a 91% reduction in the number of worried parents needing prolonged time caring for their children.

We therefore compared the first IACI’s effectiveness in the three patients who had already undergone hip surgery to the eight who had not. Results of this analysis are shown in Table 3.

Both subgroups showed positive results, confirming the reduction in analgesic use and pain episodes, and the improvement in patients’ and caregivers’ quality of life, as observed in the entire cohort. Remarkably, no significant differences were found in the rate of improvement after the first IACI when comparing preoperative and postoperative patients.

Overall, the mean reported duration of the IACIs’ benefit was 5.4 ± 2.4 months (range 1–9). Mann-Whitney’s test revealed a positive correlation between the number of IACIs and the benefit duration (r = 0.48; *p*-value = 0.04). Moreover, we found a negative correlation between age at first injection and benefit duration (r = −0.71; *p*-value = 0.02). After dividing the population into two age groups, the benefit duration of the first IACI was significantly higher in those younger than 14 years compared to those who were older (*p*-value = 0.01). No significant differences in benefit duration were shown when comparing patients who had already received hip surgery to those who had not (*p*-value = 0.517).

One child (9.1%) developed mild local swelling for two days. No other local or systemic adverse events were reported.

## 4. Discussion

Several studies have shown how hip surveillance programs improve the outcome of HD in terms of rate of dislocated hips and salvage surgery [18,34,35]. However, this strategy may be difficult to pursue in some cases given the fragility and/or the limited life expectancy of many of these patients [36], the need for invasive surgery even in asymptomatic children with the possibility of parental refusal, the high rate of complications after hip surgery [37,38,39,40], and the frequent need for repeated surgeries before reaching skeletal maturity [31]. Moreover, an older age at the time of reconstructive surgery is associated with a reduced need for surgical revision [41]. Taken together, these factors highlight the need for additional less invasive strategies to treat painful hip dislocation and postpone surgery as long as possible.

To our knowledge, only one previous retrospective study [33] evaluated the caregivers’ satisfaction with the IACI among four palliative procedures for painful HD. In this cohort 15 children with painful hip dislocation aged 8–17 years old (mean 15.5) received an average of two fluoroscopic-guided hip injections of betamethasone, with a reported significative reduction in pain (rated on a visual analog scale-11) from 7.88 to 3.08, and a treatment satisfaction score (rated by parents and caregivers on a scale from 0 to 10) of 6.83. The mean duration of the reported benefit was four months (range 2–8), but the effectiveness decreased during follow up.

Our study confirms that the IACI is an effective and safe procedure in neurologic HD, with a significant rate of resolution of pain, reduction in analgesic use, and improvement of both patients’ and caregivers’ quality of life. These effects lasted on average more than five months in our cohort. Interestingly, our results suggest that the procedure may be repeated as needed in order to obtain progressively more extended periods of wellness. The benefit seems to last longer when the IACI is performed in younger children. No significant differences were found when comparing preoperative and postoperative patients.

Even when the total number of injections was small, no relevant adverse events were reported in our cohort, including those typically associated with IACIs, such as joint erythema, bruising, skin or subcutaneous atrophy, and depigmentation [32]. These results suggest that fluoroscopy guidance may guarantee a high injection accuracy in order to avoid extravasation of drugs injected into the joint cavity, and should therefore be part of the procedure. On the other hand, given the local administration of glucocorticoids with low systemic absorption, systemic adverse events, such as Cushingoid habitus, an acne-like rash, and sleep disorders, are rarely observed following IACIs [42,43]. A possible concern is the potential detrimental effect of intra-articular glucocorticoids on articular cartilage as a result of the inhibition of the anabolic activity of chondrocytes, the decreased collagen expression, and/or a potential chondrotoxicity [44]. As an example, a randomized controlled trial [45] comparing intra-articular triamcinolone and saline in 119 adult patients with knee osteoarthritis showed a significantly greater cartilage volume loss in the triamcinolone group (−0.21 mm over 2 years of three-monthly injections vs. −0.10 mm in the saline group). However, evidence regarding the effect of intra-articular glucocorticoids on cartilage in children is scarce, and mostly limited to those affected by chronic arthritis [46], where glucocorticoids are definitely considered beneficial because of their anti-inflammatory effect. Further studies are warranted to address this issue, and to possibly establish if a safe interval before repeating injections exists. If a detrimental effect was to be confirmed, a case-by-case decision should be made in order to optimize the cost-benefit ratio in a setting where life expectancy may be limited and severity of symptoms may profoundly impact the children’s quality of life.

According to the American College of Rheumatology, an IACI is the first-line therapy for adults with rheumatoid arthritis or hip osteoarthritis, and children with oligoarticular JIA [32]. Although due to mechanical damage, hip displacement-related osteoarthritis may share a secondary inflammatory component with those conditions, so it is conceivable that the anti-inflammatory effect of the glucocorticoids is the key to understanding the observed benefit over the pain produced by hip dislocation as well.

The most commonly used drugs for IACIs in JIA include methylprednisolone acetate (MPA), triamcinolone acetonide (TA), and triamcinolone hexacetonide (TH). While in vitro studies suggest lower chondrotoxicity for MPA (with betamethasone being the most toxic) [47], TH has been shown to produce better control of symptoms and longer periods of remission when compared to TA in JIA patients [48,49,50]. Further research should address this issue in order to establish the safest and most effective drug for neurologic HD.

Taking into account the widely positive results concerning the IACI’s effectiveness, the main strengths of our study are the wide age range of our cohort, the inclusion of patients who had already received hip surgery (although no significant differences were shown when compared to those who had not), the absence of relevant local or systemic adverse events, and the prolonged study period, with an average follow up time of more than 2 years (and a maximum of approximately 4 years) from the first IACI.

The primary study’s limits are the retrospective design, the small sample size with different etiologies, a potential recall bias, and the lack of specific pain scales, leading to a reliance on the parents’ judgement. However, the literature suggests that these children’s parents are to be considered reliable in estimating their pain [51]. Moreover, all of the patients in our cohort were classified into GMFCS level V and presented hip dislocation. Such a grade of severity reflects the current contest of application of the IACI, but prevents the generalizability of our results to lower grades of motor impairment and hip displacement.

Finally, we believe that these data significantly add to the limited available evidence, allowing physicians to consider this option in a pragmatic and palliative perspective. Future randomized controlled trials, including the need for a sham sedation procedure, are needed to definitively confirm these results and clarify factors predictive of the benefit, size, duration, long-term effectiveness and adverse events.

## 5. Conclusions

This study is further confirmation of the possible role of IACIs in treating painful hip dislocation in children and young adults with neurologic impairment, showing a significant rate of resolution of pain, reduction in analgesic use, and improvement of both patients’ and caregivers’ quality of life. The IACI can be performed both before and after surgery, with the expectation of similar benefits. Younger children seem to benefit for longer from this procedure. Even if symptoms can be expected to recur or slowly progress, the IACI may be considered a test to determine which patients may benefit the most from surgical reconstruction or, in some cases, avoid surgery. No relevant local or systemic adverse events were reported in our cohort. We believe our data support the role of the IACI as a possible bridge for reconstruction surgery, or in a palliative perspective for patients in which surgery is not feasible.

## Figures and Tables

**Figure 1 children-10-01353-f001:**
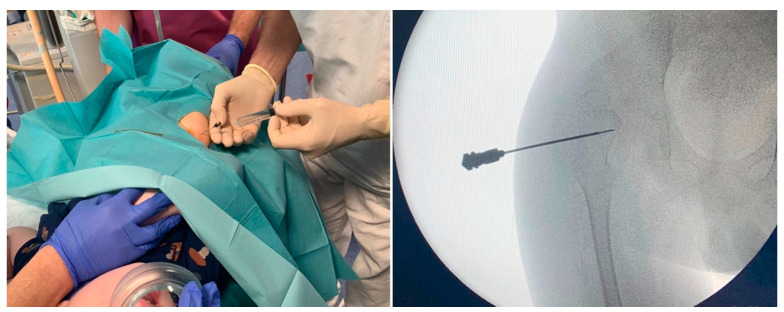
Fluoroscopic-guided hip arthrocentesis and injection of methylprednisolone acetate 40 mg plus 3 mL of 1% mepivacaine, under procedural sedation with intravenous ketamine.

**Table 1 children-10-01353-t001:** The IACIs’ timeframe in patients receiving multiple injections.

Patient	Time from 1st IACIs (Months)				
	2nd IACI	3rd IACI	4th IACI	5th IACI	6th IACI
1	12	-	-	-	-
2	13	-	-	-	-
3	11	-	-	-	-
4	2	8	-	-	-
5	6	10	16	28	32

**Table 2 children-10-01353-t002:** The effectiveness of the first IACI for pain control in 11 children with neurologic impairment suffering from hip dislocation.

	Before IACI (%)	After IACI (%)	*p*-Value
**Medical therapies**			
Use	10 (90.1)	4 (36.4)	0.02
Type			
Paracetamol	10 (100)	2 (50)	0.002
NSAIDs	7 (70)	2 (50)	0.08
Opioids	2 (20)	-	0.48
Gabapentin	2 (20)	1 (25)	1.00
Baclofen	1 (10)	1 (25)	1.00
BoNT-A	1 (10)	-	1.00
Frequency			
More than daily	5 (50)	-	0.03
Daily	-	1 (25)	1.00
2–3/week	5 (50)	-	0.03
Weekly	-	1 (25)	1.00
Monthly/occasionally	-	2 (50)	0.48
Response to treatment			
No	8 (80)	-	0.015
Yes	2 (20)	4 (100)	
**Non-medical therapy** (i.e., physiotherapy)			
Use	9 (81.8)	9 (81.8)	1.00
Frequency			
More than daily	-	-	1.00
Daily	1 (11.1)	1 (11.1)	1.00
2–3/week	6 (66.7)	6 (66.7)	1.00
Weekly	2 (22.2)	2 (22.2)	1.00
Monthly/occasionally	-	-	1.00
Response to treatment			
No	7 (77.8)	5 (55.6)	0.62
Yes	2 (22.2)	4 (44.4)	
**Pain episodes**			
Presence	11 (100)	2 (18.2)	0.0002
Frequency			
More than daily	11 (100)	1 (50)	<0.0001
Daily	-	1 (50)	1.00
2–3/week	-	-	1.00
Weekly	-	-	1.00
Monthly/occasionally	-	-	1.00
Intensity			
Mild	-	1 (50)	1.00
Moderate	-	-	1.00
Severe	9 (81.8)	1 (50)	0.002
Very severe	2 (18.2)	-	0.48
**Hip mobility**			
Mobile lower limbs	-	5 (45.4)	0.03
Passively mobile lower limbs	-	2 (18.2)	0.48
Stiff lower limbs	11 (100)	4 (36.4)	0.004
**Decubitus**			
Indifferent	-	7 (63.6)	0.004
Preferential	1 (9.1)	3 (27.3)	0.59
Obligated	10 (90.9)	1 (9.1)	0.0003
**Behavior**			
Normal, non-lamenting	1 (9.1)	9 (81.8)	0.002
Occasionally lamenting	3 (27.3)	2 (18.2)	1.00
Often lamenting	1 (9.1)	-	1.00
Occasionally crying	5 (45.4)	-	0.03
Often crying	1 (9.1)	-	1.00
**Sleep quality**			
Sleeps all night long	3 (27.3)	10 (90.9)	0.007
1–2 awakenings/night	3 (27.3)	-	0.21
≥3 awakenings/night	4 (36.3)	1 (9.1)	0.31
Not sleeping at all	1 (9.1)	-	1.00
**Appetite**			
Normal	7 (63.6)	11 (100)	0.09
Reduced	4 (36.4)	-	
**Emergency department admissions for pain**	1 (9.1)	-	1.00
**Caregivers’ quality of life**			
Not worried, normal time for care	-	10 (90.9)	<0.0001
Not worried, prolonged time for care	-	-	
Worried, normal time for care	-	-	
Worried, prolonged time for care	11 (100)	1 (9.1)	

NSAIDs = Non-steroidal anti-inflammatory drugs; BoNT-A = Botulinum neurotoxin type A.

**Table 3 children-10-01353-t003:** Comparison of the first IACI’s effectiveness in preoperative and postoperative children.

IACI Effectiveness	Preoperative Children (no. = 8)		Postoperative Children (no. = 3)	
	Before IACI	After IACI	Before IACI	After IACI
**Pain**				
*No. children using analgesics*	7	2	3	2
*No. children experiencing pain*	8	6	3	0
**Children’s quality of life**				
*No. children experiencing hip stiffness*	8	4	3	0
*No. children with obligated decubitus*	7	6	3	0
*No. children lamenting or crying*	7	4	3	0
*No. children awakening at least once a night*	7	1	3	0
**Caregivers’ quality of life**				
*No. worried parents needing prolonged time for care*	7	6	3	0

## Data Availability

The data presented in this study are available on request from the corresponding author. The data are not publicly available due to privacy reasons.

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
