# Peer review of "Study Confirms Safety and Effectiveness of Intra-Articular Glucocorticoids for Painful Hip Dislocation in Children and Young Adults with Neurologic Impairment"

_children, 2023, doi:10.3390/children10081353_

Round 1

Reviewer 1 Report

Dear Author, 

Thank you for the opportunity to review this article.

The introduction is adequate in means of elaboration.

In the Materials and Methods, you need to mention inclusion and exclusion criteria.

In the Results, what is the time lapse between the two evaluations? For how long was the follow-up included in the research?

In the Discussion you should point out the strengths and the weaknesses of your study.

Please rewrite the conclusions clearly. “Further studies...” should be at the end of the Discussion, not the Conclusions.

Reviewer 2 Report

Thank you for submitting your study on the use of intra-articular steroid injections of the hip joint in 11 children and adolescents with hip dysplasia in a variety of diagnoses. It was interesting but provided no new or useful information. As a consequence, I have concerns regarding publication. I have a number of questions, concerns and editorial recommendations to be addressed if a revision is requested.

Abstract

(1).  Page 1, Line 26.  You need to separate your patients to those that were preoperative and postoperative. Also, did these have hip dysplasia, subluxation and dislocation? It is important that your patients be appropriately categorized to be accurately assessed.

Introduction

(2). Page 1, Line 33. The Introduction is much too long. and needs to be condensed. Focus on the issues involving your study (diagnoses, classification of hip displacement., GMFCS, treatment options including steroid injections, and results. The main point is who needs steroids and the published results.

Materials and Methods

(3). Page 3, Line 118. Again, you need to divide your patients into appropriate groups of dysplasia, subluxation and dislocation and analyze the results accordingly. You may not have enough patients to perform an accurate analysis. This is a major weakness of your study.

(4). Page 3, Line 144. Was there a specific postoperative protocol, such as range of motion, stretching exercises, etc? Please make some sort of statement.

Results

(5). Page 4, Line 161. Since this is a pediatric study I would recommend that all patients 19 years of age and older be excluded.

(6). Page 4, Line 162. Please list the number of patients and treatments prior to the use of percentages. This makes your results easier to comprehend.

Conclusions

(7). Page 8, Line 262. What are your true conclusions? You need to consider the deterioration of articular cartilage with repetitive steroids injections. In view of this I feel your results are primarily limited to transient problems and as test for possible preoperative planning for surgical reconstruction. The transient improvement is already well known. The duration of the affects are limited to the severity of hip dysplasia. This is why this is an important issue.

The quality of the English is satisfactory but can be improved.

Reviewer 3 Report

Dear Authors,

An interesting topic is highlighted in this study; the effectiveness of intra-articular injections in paediatric hip displacement due to neurological deficiency. As far as I am considered, some points regarding method, design and results of this study needing further explanation and clarification. The major possible fault is the lack of any justification of the total of injections per participant, the small number of participants and the absence of any power analysis (type II error could be significant high due to small participant number) and the retrospective design of the interview adds a significant amount of bias - any caregiver is rarely objective especially when recalling facts from the past. 

Line 3: Children; Participants of the study are both children and adolescents or even young adults. Please be exact.

Line 13: Please clarify the meaning of this phrase. Any reader should understand why a surgical treatment could not be a permanent solution after a conservative treatment failure. This delay of operative treatment could negatively affect the final result?

Line 16: Instead of children use something "participants" or "patients" (see above)

Lines 18-19: A relative small number of participants and a big range of their ages. A power analysis is eminent to be reached scientific soundness of this study. Moreover, use terms like reduce or reduction instead of symbols (-). 

Line 32: This section is big and mostly irrelevant to the subject. Please exclude the first three paragraphs and focus on facts related to clinical condition of patients with neurologic impairment and hip dislocation/ displacement.

Line 117: Please provide inclusion and exclusion criteria. Moreover, power analysis should be added in this point. Are this study data reliable and statistically significant? Were these participants the unique patients of the center or were also some losses?

Line 131: Orthopaedic surgeon instead of orthopedist.

Line 132: Please provide data regarding the "specific expertise".

Lines 160-161: Any person older than 16 is not classified as child. 

Lines 161-163: Please provide data regarding the absolute number of participants with the exact neurological impairment. It is also important the heterogeneity of the sample to be calculated (I2). The statistical base of this study should be proven.

Lines 166-169: Please provide data for all possible surgical treatments of participants. Was there any hardware inside joint capsule?

Line 174-177: Please clarify the treatment algorithm. Which was the frequency of the injections? According to which clinical data? Which was the follow-up time and which the end time?

Table 1: Instead of inferior limbs , "lower Limbs"

Lines 186-207: Please be more thorough. Provide more explanations and avoid the use of symbols (-).  Make correlations between different clinical conditions or compare surgical techniques and results. Mention the time between index surgery and injection. When this injection is using preoperatively, which are the results of the surgery? Was the follow-up time sufficient?

LIne 210: Please expand this section. Provide a more thorough aspect of this topic and its significance according to literature.

Lines 231-232: Please provide the time frame of these side effects

Lines 233-234: Please clarify this correlation between patient age and injection benefits.

Round 2

Reviewer 2 Report

Thank you for revising your manuscript on intra-articular glucocorticoid injections for painful hips in patients with neurologic disorders and either subluxation or dislocation of their hips. I have again reviewed your manuscript and I still do not feel you adequately addressed my previous questions or concerns. At best your technique offers transient improvement in hip symptoms in these patients. It is not definitive treatment. My concerns remain:

(1). The small number of patients (11 patients) and involved hips (19 hips) with multiple diagnoses. You cannot make any significant conclusions on such small numbers. Because the etiology was different and in several major diagnostic groups this makes comparisons even more difficult.

(2). The failure to specify whether the hips were subluxated or dislocated. Subluxation means there was still contact between the femoral head and acetabulum but abnormally related while dislocation implies no contact. It is well-known that this is an important distinction. I would anticpate more symptoms and less response to steroid injections in the subluxations due to more articular cartilage degeneration (osteoarthritis).

(3). All patients were listed as non-ambulatory (GMFCS V) and with Reimer's migration index >80%. This still does not define subluxation or dislocation.

(4). The short duration of improvement of a mean of 5.4+/- 2.4 months. Repeated injections are not the answer in this setting. Hip reconstruction surgery should offer lasting improvement.

(5). The conclusions, in my opinion, were limited. I do not accept the statement that "These data suggest a long-lasting benefit which seems to increase as multiple intra-articular corticosteroid injections are performed in the same child". The fact that younger children did better than those who were older is not surprising and probably represents advancing degenerative osteoarthritis. At best your procedure is a test to determine who may benefit from surgical hip reconstruction. 

The English has been improved and is now satisfactory. A review by an scientific English editor would be helpful from a scientific perspective.

Reviewer 3 Report

Dear authors,

All points were finely addressed. Few  points raised: the use of minus symbol  before every percentage regarding reduction. Please remove it. Moreover, in line 21 word "motility" could preferable be "mobility". Last but not at least (lines 149-150) you should be more precise regarding the indications for a second or more injections. This study proves the effectiveness of IACI regarding the quality of life of both patients and their caregivers. Timetable and timeframe for each patient regarding number of injections and time in-between is lacking. So you should either clarify and justify the firm indications for another IACI or form another table with the timeline of injection and possible other interventions (operative treatment etc) for every patient. Then all concerns for IACI risks (chondrocytes degeneration, infection etc) outweigh possible benefits would be erased. 

Round 3

Reviewer 2 Report

Your manuscript is improved. I feel you are now grasping my concerns. I feel your manuscript has potential but only as a temporizing procedure. My current concerns include:

(1). Page 3, Line 132. Did any patients receive BoNT-A after any IACI? If so these patients need to be excluded as they received mixed treatment.

(2). Page 5, Line 194. Reimer’s Migration Index >80% still allowed some patients to have contact between the margin of the acetabulum and femoral head. This would produce high contact pressures, especially when coupled with spasticity, hastening degenerative osteoarthritis. When there is no contact between the acetabulum and femoral head due to interposition of the capsule  and the muscles of the ilium the pressure is usually less and the symptoms less intense. This is why a more accurate distinction between subluxation and dislocation is necessary.

(3). Page 9, Line 305,. This statement is not comparable to your report. Rheumatoid Arthritis is an inflammatory condition involving the synovium while your cases have degenerative osteoarthritis due to abnormalities of joint alignment (hip subluxation or dislocation) and mechanical degeneration.

(4). Page 19, Line 340. You conclusions are improved. Your procedure is temporizing, at best, but not definitive. It could also be considered in medically fragile patients in which definitive hip surgery may have high risks. However, I would assume the symptoms will persist and slowly progress.
